# Model-Based Imitation Learning for Urban Driving

**Anthony Hu[1,2]**  **Gianluca Corrado[1]**  **Nicolas Griffiths[1]**  **Zak Murez[1]**  **Corina Gurau[1]**

**Hudson Yeo[1]**  **Alex Kendall[1]**  **Roberto Cipolla[2]**  **Jamie Shotton[1]**

[1]Wayve, UK.  [2]University of Cambridge, UK.
research@wayve.ai

## Abstract

An accurate model of the environment and the dynamic agents acting in it offers great potential for improving motion planning. We present MILE: a Model-based Imitation LEarning approach to jointly learn a model of the world and a policy for autonomous driving. Our method leverages 3D geometry as an inductive bias and learns a highly compact latent space directly from high-resolution videos of expert demonstrations. Our model is trained on an offline corpus of urban driving data, without any online interaction with the environment. MILE improves upon prior state-of-the-art by 31% in driving score on the CARLA simulator when deployed in a completely new town and new weather conditions. Our model can predict diverse and plausible states and actions, that can be interpretably decoded to bird's-eye view semantic segmentation. Further, we demonstrate that it can execute complex driving manoeuvres from plans entirely predicted in imagination. Our approach is the first camera-only method that models static scene, dynamic scene, and ego-behaviour in an urban driving environment. The code and model weights are available at https://github.com/wayveai/mile.

## 1  Introduction

From an early age we start building internal representations of the world through observation and interaction [1]. Our ability to estimate scene geometry and dynamics is paramount to generating complex and adaptable movements. This accumulated knowledge of the world, part of what we often refer to as common sense, allows us to navigate effectively in unfamiliar situations [2].

In this work, we present MILE, a Model-based Imitation LEarning approach to jointly learn a model of the world and a driving policy. We demonstrate the effectiveness of our approach in the autonomous driving domain, operating on complex visual inputs labelled only with expert action and semantic segmentation. Unlike prior work on world models [3, 4, 5], our method does not assume access to a ground truth reward, nor does it need any online interaction with the environment. Further, previous environments in OpenAI Gym [3], MuJoCo [4], and Atari [5] were characterised by simplified visual inputs as small as $64 \times 64$ images. In contrast, MILE operates on high-resolution camera observations of urban driving scenes.

Driving inherently requires a geometric understanding of the environment, and MILE exploits 3D geometry as an important inductive bias by first lifting image features to 3D and pooling them into a bird's-eye view (BeV) representation. The evolution of the world is modelled by a latent dynamics model that infers compact latent states from observations and expert actions. The learned latent state is the input to a driving policy that outputs vehicle control, and can additionally be decoded to BeV segmentation for visualisation and as a supervision signal.

36th Conference on Neural Information Processing Systems (NeurIPS 2022).

Our method also relaxes the assumption made in some recent work [6, 7] that neither the agent nor its actions influence the environment. This assumption rarely holds in urban driving, and therefore MILE is action-conditioned, allowing us to model how other agents respond to ego-actions. We show that our model can predict plausible and diverse futures from latent states and actions over long time horizons. It can even predict entire driving plans in imagination to successfully execute complex driving manoeuvres, such as negotiating a roundabout, or swerving to avoid a motorcyclist (see videos in the supplementary material).

We showcase the performance of our model on the driving simulator CARLA [8], and demonstrate a new state-of-the-art. MILE achieves a 31% improvement in driving score with respect to previous methods [9, 10] when tested in a new town and new weather conditions. Finally, during inference, because we model time with a recurrent neural network, we can maintain a single state that summarises all the past observations and then efficiently update the state when a new observation is available. We demonstrate that this design decision has important benefits for deployment in terms of latency, with negligible impact on the driving performance.

To summarise the main contributions of this paper:

- We introduce a novel model-based imitation learning architecture that scales to the visual complexity of autonomous driving in urban environments by leveraging 3D geometry as an inductive bias. Our method is trained solely using an offline corpus of expert driving data, and does not require any interaction with an online environment or access to a reward, offering strong potential for real-world application.

- Our camera-only model sets a new state-of-the-art on the CARLA simulator, surpassing other approaches, including those requiring LiDAR inputs.

- Our model predicts a distribution of diverse and plausible futures states and actions. We demonstrate that it can execute complex driving manoeuvres from plans entirely predicted in imagination.

## 2   Related Work

Our work is at the intersection of imitation learning, 3D scene representation, and world modelling.

**Imitation learning.**   Despite that the first end-to-end method for autonomous driving was envisioned more than 30 years ago [11], early autonomous driving approaches were dominated by modular frameworks, where each module solves a specific task [12, 13, 14]. Recent years have seen the development of several end-to-end self-driving systems that show strong potential to improve driving performance by predicting driving commands from high-dimensional observations alone. Conditional imitation learning has proven to be one successful method to learn end-to-end driving policies that can be deployed in simulation [15] and real-world urban driving scenarios [16]. Nevertheless, difficulties of learning end-to-end policies from high-dimensional visual observations and expert trajectories alone have been highlighted [17].

Several works have attempted to overcome such difficulties by moving past pure imitation learning. DAgger [18] proposes iterative dataset aggregation to collect data from trajectories that are likely to be experienced by the policy during deployment. NEAT [19] additionally supervises the model with BeV semantic segmentation. ChauffeurNet [20] exposes the learner to synthesised perturbations of the expert data in order to produce more robust driving policies. Learning from All Vehicles (LAV) [10] boosts sample efficiency by learning behaviours from not only the ego vehicle, but from all the vehicles in the scene. Roach [9] presents an agent trained with supervision from a reinforcement learning coach that was trained on-policy and with access to privileged information.

**3D scene representation.**   Successful planning for autonomous driving requires being able to understand and reason about the 3D scene, and this can be challenging from monocular cameras. One common solution is to condense the information from multiple cameras to a single bird's-eye representation of the scene. This can be achieved by lifting each image in 3D (by learning a depth distribution of features) and then splatting all frustums into a common rasterised BeV grid [21, 22, 23]. An alternative approach is to rely on transformers to learn the direct mapping from image to bird's-eye view [24, 25, 26], without explicitly modelling depth.

**World models.** Model-based methods have mostly been explored in a reinforcement learning setting and have been shown to be extremely successful [3, 27, 5]. These methods assume access to a reward, and online interaction with the environment, although progress has been made on fully offline reinforcement learning [28, 29]. Model-based imitation learning has emerged as an alternative to reinforcement learning in robotic manipulation [30] and OpenAI Gym [31]. Even though these methods do not require access to a reward, they still require online interaction with the environment to achieve good performance.

Learning the latent dynamics of a world model from image observations was first introduced in video prediction [32, 33, 34]. Most similar to our approach, [4, 5] additionally modelled the reward function and optimised a policy inside their world model. Contrarily to prior work, our method does not assume access to a reward function, and directly learns a policy from an offline dataset. Additionally, previous methods operate on simple visual inputs, mostly of size $64 \times 64$. In contrast, MILE is able to learn the latent dynamics of complex urban driving scenes from high resolution $600 \times 960$ input observations, which is important to ensure small details such as traffic lights can be perceived reliably.

**Trajectory forecasting.** The goal of trajectory forecasting is to estimate the future trajectories of dynamic agents using past physical states (e.g. position, velocity), and scene context (e.g. as an offline HD map) [35, 36, 37, 38]. World models build a latent representation of the environment that explains the observations from the sensory inputs of the ego-agent (e.g. camera images) conditioned on their actions. While trajectory forecasting methods only model the dynamic scene, world models jointly reason on static and dynamic scenes. The future trajectories of moving agents is implicitly encoded in the learned latent representation of the world model, and could be explicitly decoded given we have access to future trajectory labels.

[35, 37, 38] forecast the future trajectory of moving agents, but did not control the ego-agent. They focused on the prediction problem and not on learning expert behaviour from demonstrations. [39] inferred future trajectories of the ego-agent from expert demonstrations, and conditioned on some specified goal to perform new tasks. [36] extended their work to jointly model the future trajectories of moving agents as well as of the ego-agent.

Our proposed model jointly models the motion of other dynamics agents, the behaviour of the ego-agent, as well as the static scene. Contrary to prior work, we do not assume access to ground truth physical states (position, velocity) or to an offline HD map for scene context. Our approach is the first camera-only method that models static scene, dynamic scene, and ego-behaviour in an urban driving environment.

## 3 MILE: Model-based Imitation LEarning

In this section, we present MILE: our method that learns to jointly control an autonomous vehicle and model the world and its dynamics. An overview of the architecture is presented in Figure 1 and the full description of the network can be found in Appendix C. We begin by defining the generative model (Section 3.1), and then derive the inference model (Section 3.2). Section 3.3 and Section 3.4 describe the neural networks that parametrise the inference and generative models respectively. Finally, in Section 3.5 we show how our model can predict future states and actions to drive in imagination.

### 3.1 Probabilistic Generative Model

Let $\mathbf{o}_{1:T}$ be a sequence of $T$ video frames with associated expert actions $\mathbf{a}_{1:T}$ and ground truth BeV semantic segmentation labels $\mathbf{y}_{1:T}$. We model their evolution by introducing latent variables $\mathbf{s}_{1:T}$ that govern the temporal dynamics. The initial distribution is parameterised as $\mathbf{s}_1 \sim \mathcal{N}(\mathbf{0}, \boldsymbol{I})$, and we additionally introduce a variable $\mathbf{h}_1 \sim \delta(\mathbf{0})$ that serves as a deterministic history. The transition consists of (i) a deterministic update $\mathbf{h}_{t+1} = f_\theta(\mathbf{h}_t, \mathbf{s}_t)$ that depends on the past history $\mathbf{h}_t$ and past state $\mathbf{s}_t$, followed by (ii) a stochastic update $\mathbf{s}_{t+1} \sim \mathcal{N}(\mu_\theta(\mathbf{h}_{t+1}, \mathbf{a}_t), \sigma_\theta(\mathbf{h}_{t+1}, \mathbf{a}_t)\boldsymbol{I})$, where we parameterised $\mathbf{s}_t$ as a normal distribution with diagonal covariance. We model these transitions with neural networks: $f_\theta$ is a gated recurrent cell, and $(\mu_\theta, \sigma_\theta)$ are multi-layer perceptrons. The full probabilistic model is given by Equation (1).

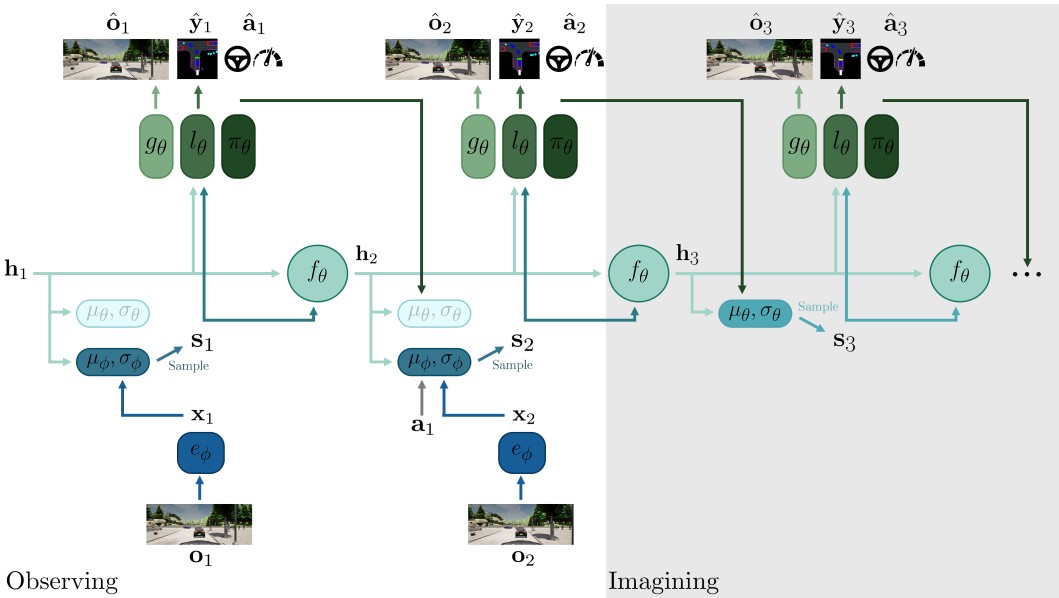

Figure 1: Architecture of MILE.

(i) The goal is to infer the **latent dynamics** $(\mathbf{h}_{1:T}, \mathbf{s}_{1:T})$ that generated the observations $\mathbf{o}_{1:T}$, the expert actions $\mathbf{a}_{1:T}$ and the bird's-eye view labels $\mathbf{y}_{1:T}$. The latent dynamics contains a deterministic history $\mathbf{h}_t$ and a stochastic state $\mathbf{s}_t$.

(ii) The **inference model**, with parameters $\phi$, estimates the posterior distribution of the stochastic state $q(\mathbf{s}_t|\mathbf{o}_{\leq t}, \mathbf{a}_{<t}) \sim \mathcal{N}(\mu_\phi(\mathbf{h}_t, \mathbf{a}_{t-1}, \mathbf{x}_t), \sigma_\phi(\mathbf{h}_t, \mathbf{a}_{t-1}, \mathbf{x}_t)\boldsymbol{I})$ with $\mathbf{x}_t = e_\phi(\mathbf{o}_t)$. $e_\phi$ is the observation encoder that lifts image features to 3D, pools them to bird's-eye view, and compresses to a 1D vector.

(iii) The **generative model**, with parameters $\theta$, estimates the prior distribution of the stochastic state $p(\mathbf{s}_t|\mathbf{h}_{t-1}, \mathbf{s}_{t-1}) \sim \mathcal{N}(\mu_\theta(\mathbf{h}_t, \hat{\mathbf{a}}_{t-1}), \sigma_\theta(\mathbf{h}_t, \hat{\mathbf{a}}_{t-1})\boldsymbol{I})$, with $\mathbf{h}_t = f_\theta(\mathbf{h}_{t-1}, \mathbf{s}_{t-1})$ the deterministic transition, and $\hat{\mathbf{a}}_{t-1} = \pi_\theta(\mathbf{h}_{t-1}, \mathbf{s}_{t-1})$ the predicted action. It additionally estimates the distributions of the observation $p(\mathbf{o}_t|\mathbf{h}_t, \mathbf{s}_t) \sim \mathcal{N}(g_\theta(\mathbf{h}_t, \mathbf{s}_t), \boldsymbol{I})$, the bird's-eye view segmentation $p(\mathbf{y}_t|\mathbf{h}_t, \mathbf{s}_t) \sim \mathrm{Categorical}(l_\theta(\mathbf{h}_t, \mathbf{s}_t))$, and the action $p(\mathbf{a}_t|\mathbf{h}_t, \mathbf{s}_t) \sim \mathrm{Laplace}(\pi_\theta(\mathbf{h}_t, \mathbf{s}_t), \mathbf{1})$.

(iv) In the diagram, we represented our model observing inputs for $T = 2$ timesteps, and then imagining future latent states and actions for one step.

$$
\begin{cases}
\mathbf{h}_1 & \sim \delta(\mathbf{0}) \\
\mathbf{s}_1 & \sim \mathcal{N}(\mathbf{0}, \boldsymbol{I}) \\
\mathbf{h}_{t+1} & = f_\theta(\mathbf{h}_t, \mathbf{s}_t) \\
\mathbf{s}_{t+1} & \sim \mathcal{N}(\mu_\theta(\mathbf{h}_{t+1}, \mathbf{a}_t), \sigma_\theta(\mathbf{h}_{t+1}, \mathbf{a}_t)\boldsymbol{I}) \\
\mathbf{o}_t & \sim \mathcal{N}(g_\theta(\mathbf{h}_t, \mathbf{s}_t), \boldsymbol{I}) \\
\mathbf{y}_t & \sim \mathrm{Categorical}(l_\theta(\mathbf{h}_t, \mathbf{s}_t)) \\
\mathbf{a}_t & \sim \mathrm{Laplace}(\pi_\theta(\mathbf{h}_t, \mathbf{s}_t), \mathbf{1})
\end{cases}
\tag{1}
$$

with $\delta$ the Dirac delta function, $g_\theta$ the image decoder, $l_\theta$ the BeV decoder, and $\pi_\theta$ the policy, which will be described in Section 3.4.

## 3.2 Variational Inference

Following the generative model described in Equation (1), we can factorise the joint probability as:

$$
p(\mathbf{o}_{1:T}, \mathbf{y}_{1:T}, \mathbf{a}_{1:T}, \mathbf{h}_{1:T}, \mathbf{s}_{1:T}) = \prod_{t=1}^{T} p(\mathbf{h}_t, \mathbf{s}_t|\mathbf{h}_{t-1}, \mathbf{s}_{t-1}, \mathbf{a}_{t-1}) p(\mathbf{o}_t, \mathbf{y}_t, \mathbf{a}_t|\mathbf{h}_t, \mathbf{s}_t)
\tag{2}
$$

with

$$p(\mathbf{h}_t, \mathbf{s}_t | \mathbf{h}_{t-1}, \mathbf{s}_{t-1}, \mathbf{a}_{t-1}) = p(\mathbf{h}_t | \mathbf{h}_{t-1}, \mathbf{s}_{t-1}) p(\mathbf{s}_t | \mathbf{h}_t, \mathbf{a}_{t-1}) \quad (3)$$

$$p(\mathbf{o}_t, \mathbf{y}_t, \mathbf{a}_t | \mathbf{h}_t, \mathbf{s}_t) = p(\mathbf{o}_t | \mathbf{h}_t, \mathbf{s}_t) p(\mathbf{y}_t | \mathbf{h}_t, \mathbf{s}_t) p(\mathbf{a}_t | \mathbf{h}_t, \mathbf{s}_t) \quad (4)$$

Given that $\mathbf{h}_t$ is deterministic according to Equation (1), we have $p(\mathbf{h}_t | \mathbf{h}_{t-1}, \mathbf{s}_{t-1}) = \delta(\mathbf{h}_t - f_\theta(\mathbf{h}_{t-1}, \mathbf{s}_{t-1}))$. Therefore, in order to maximise the marginal likelihood of the observed data $p(\mathbf{o}_{1:T}, \mathbf{y}_{1:T}, \mathbf{a}_{1:T})$, we need to infer the latent variables $\mathbf{s}_{1:T}$. We do this through deep variational inference by introducing a variational distribution $q_{H,S}$ defined and factorised as follows:

$$q_{H,S} \triangleq q(\mathbf{h}_{1:T}, \mathbf{s}_{1:T} | \mathbf{o}_{1:T}, \mathbf{a}_{1:T-1}) = \prod_{t=1}^{T} q(\mathbf{h}_t | \mathbf{h}_{t-1}, \mathbf{s}_{t-1}) q(\mathbf{s}_t | \mathbf{o}_{\leq t}, \mathbf{a}_{<t}) \quad (5)$$

with $q(\mathbf{h}_t | \mathbf{h}_{t-1}, \mathbf{s}_{t-1}) = p(\mathbf{h}_t | \mathbf{h}_{t-1}, \mathbf{s}_{t-1})$, the Delta dirac function defined above, and $q(\mathbf{h}_1) = \delta(\mathbf{0})$. We parameterise this variational distribution with a neural network with weights $\phi$. By applying Jensen's inequality, we can obtain a variational lower bound on the log evidence:

$$\log p(\mathbf{o}_{1:T}, \mathbf{y}_{1:T}, \mathbf{a}_{1:T}) \geq \mathcal{L}(\mathbf{o}_{1:T}, \mathbf{y}_{1:T}, \mathbf{a}_{1:T}; \theta, \phi)$$

$$\triangleq \sum_{t=1}^{T} \mathbb{E}_{q(\mathbf{h}_{1:t}, \mathbf{s}_{1:t} | \mathbf{o}_{\leq t}, \mathbf{a}_{<t})} \Big[ \underbrace{\log p(\mathbf{o}_t | \mathbf{h}_t, \mathbf{s}_t)}_{\text{image reconstruction}} + \underbrace{\log p(\mathbf{y}_t | \mathbf{h}_t, \mathbf{s}_t)}_{\text{bird's-eye segmentation}} + \underbrace{\log p(\mathbf{a}_t | \mathbf{h}_t, \mathbf{s}_t)}_{\text{action}} \Big]$$

$$- \sum_{t=1}^{T} \mathbb{E}_{q(\mathbf{h}_{1:t-1}, \mathbf{s}_{1:t-1} | \mathbf{o}_{\leq t-1}, \mathbf{a}_{<t-1})} \Big[ \underbrace{D_{\mathrm{KL}}\Big( q(\mathbf{s}_t | \mathbf{o}_{\leq t}, \mathbf{a}_{<t}) \, \| \, p(\mathbf{s}_t | \mathbf{h}_{t-1}, \mathbf{s}_{t-1}) \Big)}_{\text{posterior and prior matching}} \Big] \quad (6)$$

Please refer to Appendix B for the full derivation. We model $q(\mathbf{s}_t | \mathbf{o}_{\leq t}, \mathbf{a}_{<t})$ as a Gaussian distribution so that the Kullback-Leibler (KL) divergence can be computed in closed-form. Given that the image observations $\mathbf{o}_t$ are modelled as Gaussian distributions with unit variance, the resulting loss is the mean-squared error. Similarly, the action being modelled as a Laplace distribution and the BeV labels as a categorical distribution, the resulting losses are, respectively, $L_1$ and cross-entropy. The expectations over the variational distribution can be efficiently approximated with a single sequence sample from $q_{H,S}$, and backpropagating gradients with the reparametrisation trick [40].

### 3.3 Inference Network $\phi$

The inference network, parameterised by $\phi$, models $q(\mathbf{s}_t | \mathbf{o}_{\leq t}, \mathbf{a}_{<t})$, which approximates the true posterior $p(\mathbf{s}_t | \mathbf{o}_{\leq t}, \mathbf{a}_{<t})$. It is constituted of two elements: the observation encoder $e_\phi$, that embeds input images, route map and vehicle control sensor data to a low-dimensional vector, and the posterior network $(\mu_\phi, \sigma_\phi)$, that estimates the probability distribution of the Gaussian posterior.

#### 3.3.1 Observation Encoder

The state of our model should be compact and low-dimensional in order to effectively learn dynamics. Therefore, we need to embed the high resolution input images to a low-dimensional vector. Naively encoding this image to a 1D vector similarly to an image classification task results in poor performance as shown in Section 5.2. Instead, we explicitly encode 3D geometric inductive biases in the model.

**Lifting image features to 3D.** Since autonomous driving is a geometric problem where it is necessary to reason on the static scene and dynamic agents in 3D, we first lift the image features to 3D. More precisely, we encode the image inputs $\mathbf{o}_t \in \mathbb{R}^{3 \times H \times W}$ with an image encoder to extract features $\mathbf{u}_t \in \mathbb{R}^{C_e \times H_e \times W_e}$. Then similarly to Philion and Fidler [21], we predict a depth probability distribution for each image feature along a predefined grid of depth bins $\mathbf{d}_t \in \mathbb{R}^{D \times H_e, \times W_e}$. Using the depth probability distribution, the camera intrinsics $K$ and extrinsics $M$, we can lift the image features to 3D: $\mathrm{Lift}(\mathbf{u}_t, \mathbf{d}_t, K^{-1}, M)) \in \mathbb{R}^{C_e \times D \times H_e \times D_e \times 3}$.

**Pooling to BeV.** The 3D feature voxels are then sum-pooled to BeV space using a predefined grid with spatial extent $H_b \times W_b$ and spatial resolution $b_{\mathrm{res}}$. The resulting feature is $\mathbf{b}_t \in \mathbb{R}^{C_e \times H_b \times W_b}$.

**Mapping to a 1D vector.** In traditional computer vision tasks (e.g. semantic segmentation [41], depth prediction [42]), the bottleneck feature is usually a spatial tensor, in the order of $10^5 - 10^6$ features. Such high dimensionality is prohibitive for a world model that has to match the distribution of the priors (what it thinks will happen given the executed action) to the posteriors (what actually happened by observing the image input). Therefore, using a convolutional backbone, we compress the BeV feature $\mathbf{b}_t$ to a single vector $\mathbf{x}'_t \in \mathbb{R}^{C'}$. As shown in Section 5.2, we found it critical to compress in BeV space rather than directly in image space.

**Route map and speed.** We provide the agent with a goal in the form of a route map [9], which is a small grayscale image indicating to the agent where to navigate at intersections. The route map is encoded using a convolutional module resulting in a 1D feature $\mathbf{r}_t$. The current speed is encoded with fully connected layers as $\mathbf{m}_t$. At each timestep $t$, the observation embedding $\mathbf{x}_t$ is the concatenation of the image feature, route map feature and speed feature: $\mathbf{x}_t = [\mathbf{x}'_t, \mathbf{r}_t, \mathbf{m}_t] \in \mathbb{R}^C$, with $C = 512$

### 3.3.2  Posterior Network

The posterior network $(\mu_\phi, \sigma_\phi)$ estimates the parameters of the variational distribution $q(\mathbf{s}_t|\mathbf{o}_{\leq t}, \mathbf{a}_{<t}) \sim \mathcal{N}(\mu_\phi(\mathbf{h}_t, \mathbf{a}_{t-1}, e_\phi(\mathbf{o}_t)), \sigma_\phi(\mathbf{h}_t, \mathbf{a}_{t-1}, e_\phi(\mathbf{o}_t))\boldsymbol{I})$ with $\mathbf{h}_t = f_\theta(\mathbf{h}_{t-1}, \mathbf{s}_{t-1})$. Note that $\mathbf{h}_t$ was inferred using $f_\theta$ because we have assumed that $\mathbf{h}_t$ is deterministic, meaning that $q(\mathbf{h}_t|\mathbf{h}_{t-1}, \mathbf{s}_{t-1}) = p(\mathbf{h}_t|\mathbf{h}_{t-1}, \mathbf{s}_{t-1}) = \delta(\mathbf{h}_t - f_\theta(\mathbf{h}_{t-1}, \mathbf{s}_{t-1}))$. The dimension of the Gaussian distribution is equal to 512.

### 3.4  Generative Network $\theta$

The generative network, parameterised by $\theta$, models the latent dynamics $(\mathbf{h}_{1:T}, \mathbf{s}_{1:T})$ as well as the generative process of $(\mathbf{o}_{1:T}, \mathbf{y}_{1:T}, \mathbf{a}_{1:T})$. It comprises a gated recurrent cell $f_\theta$, a prior network $(\mu_\theta, \sigma_\theta)$, an image decoder $g_\theta$, a BeV decoder $l_\theta$, and a policy $\pi_\theta$.

The prior network estimates the parameters of the Gaussian distribution $p(\mathbf{s}_t|\mathbf{h}_{t-1}, \mathbf{s}_{t-1}) \sim \mathcal{N}(\mu_\theta(\mathbf{h}_t, \hat{\mathbf{a}}_{t-1}), \sigma_\theta(\mathbf{h}_t, \hat{\mathbf{a}}_{t-1})\boldsymbol{I})$ with $\mathbf{h}_t = f_\theta(\mathbf{h}_{t-1}, \mathbf{s}_{t-1})$ and $\hat{\mathbf{a}}_{t-1} = \pi_\theta(\mathbf{h}_{t-1}, \mathbf{s}_{t-1})$. Since the prior does not have access to the ground truth action $\mathbf{a}_{t-1}$, the latter is estimated with the learned policy $\hat{\mathbf{a}}_{t-1} = \pi_\theta(\mathbf{h}_{t-1}, \mathbf{s}_{t-1})$.

The Kullback-Leibler divergence loss between the prior and posterior distributions can be interpreted as follows. Given the past state $(\mathbf{h}_{t-1}, \mathbf{s}_{t-1})$, the objective is to predict the distribution of the next state $\mathbf{s}_t$. As we model an active agent, this transition is decomposed into (i) action prediction and (ii) next state prediction. This transition estimation is compared to the posterior distribution that has access to the ground truth action $\mathbf{a}_{t-1}$, and the image observation $\mathbf{o}_t$. The prior distribution tries to match the posterior distribution. This divergence matching framework ensures the model predicts actions and future states that explain the observed data. The divergence of the posterior from the prior measures how many nats of information were missing from the prior when observing the posterior. At training convergence, the prior distribution should be able to model all action-state transitions from the expert dataset.

The image and BeV decoders have an architecture similar to StyleGAN [43]. The prediction starts as a learned constant tensor, and is progressively upsampled to the final resolution. At each resolution, the latent state is injected in the network with adaptive instance normalisation. This allows the latent states to modulate the predictions at different resolutions. The policy is a multi-layer perceptron. Please refer to Appendix C for a full description of the neural networks.

### 3.5  Imagining Future States and Actions

Our model can imagine future latent states by using the learned policy to infer actions $\hat{\mathbf{a}}_{T+i} = \pi_\theta(\mathbf{h}_{T+i}, \mathbf{s}_{T+i})$, predicting the next deterministic state $\mathbf{h}_{T+i+1} = f_\theta(\mathbf{h}_{T+i}, \mathbf{s}_{T+i})$ and sampling from the prior distribution $\mathbf{s}_{T+i+1} \sim \mathcal{N}(\mu_\theta(\mathbf{h}_{T+i+1}, \hat{\mathbf{a}}_{T+i}), \sigma_\theta(\mathbf{h}_{T+i+1}, \hat{\mathbf{a}}_{T+i})\boldsymbol{I})$, for $i \geq 0$. This process can be iteratively applied to generate sequences of longer futures in latent space, and the predicted futures can be visualised through the decoders.

Table 1: Driving performance on a new town and new weather conditions in CARLA. Metrics are averaged across three runs. We include reward signals from past work where available.

|  | Driving Score | Route | Infraction | Reward | Norm. Reward |
|---|---|---|---|---|---|
| CILRS [17] | $7.8 \pm 0.3$ | $10.3 \pm 0.0$ | $76.2 \pm 0.5$ | - | - |
| LBC [47] | $12.3 \pm 2.0$ | $31.9 \pm 2.2$ | $66.0 \pm 1.7$ | - | - |
| TransFuser [48] | $31.0 \pm 3.6$ | $47.5 \pm 5.3$ | $\mathbf{76.8 \pm 3.9}$ | - | - |
| Roach [9] | $41.6 \pm 1.8$ | $96.4 \pm 2.1$ | $43.3 \pm 2.8$ | $4236 \pm 468$ | $0.34 \pm 0.05$ |
| LAV [10] | $46.5 \pm 3.0$ | $69.8 \pm 2.3$ | $73.4 \pm 2.2$ | - | - |
| **MILE** | $\mathbf{61.1 \pm 3.2}$ | $\mathbf{97.4 \pm 0.8}$ | $63.0 \pm 3.0$ | $\mathbf{7621 \pm 460}$ | $\mathbf{0.67 \pm 0.02}$ |
| Expert | $88.4 \pm 0.9$ | $97.6 \pm 1.2$ | $90.5 \pm 1.2$ | $8694 \pm 88$ | $0.70 \pm 0.01$ |

# 4 Experimental Setting

**Dataset.** The training data was collected in the CARLA simulator with an expert reinforcement learning (RL) agent [9] that was trained using privileged information as input (BeV semantic segmentations and vehicle measurements). This RL agent generates more diverse runs and has greater driving performance than CARLA's in-built autopilot [9].

We collect data at 25Hz in four different training towns (Town01, Town03, Town04, Town06) and four weather conditions (ClearNoon, WetNoon, HardRainNoon, ClearSunset) for a total of 2.9M frames, or 32 hours of driving data. At each timestep, we save a tuple $(\mathbf{o}_t, \mathbf{route}_t, \mathbf{speed}_t, \mathbf{a}_t, \mathbf{y}_t)$, with $\mathbf{o}_t \in \mathbb{R}^{3 \times 600 \times 960}$ the forward camera RGB image, $\mathbf{route}_t \in \mathbb{R}^{1 \times 64 \times 64}$ the route map (visualized as an inset on the top right of the RGB images in Figure 2), $\mathbf{speed}_t \in \mathbb{R}$ the current velocity of the vehicle, $\mathbf{a}_t \in \mathbb{R}^2$ the action executed by the expert (acceleration and steering), and $\mathbf{y}_t \in \mathbb{R}^{C_b \times 192 \times 192}$ the BeV semantic segmentation. There are $C_b = 8$ semantic classes: background, road, lane marking, vehicles, pedestrians, and traffic light states (red, yellow, green). In urban driving environments, the dynamics of the scene do not contain high frequency components, which allows us to subsample frames at 5Hz in our sequence model.

**Training.** Our model was trained for $50,000$ iterations on a batch size of 64 on 8 V100 GPUs, with training sequence length $T = 12$. We used the AdamW optimiser [44] with learning rate $10^{-4}$ and weight decay $0.01$.

**Metrics.** We report metrics from the CARLA challenge [45] to measure on-road performance: route completion, infraction penalty, and driving score. These metrics are however very coarse, as they only give a sense of how well the agent performs with hard penalties (such as hitting virtual pedestrians). Core driving competencies such as lane keeping and driving at an appropriate speed are obscured. Therefore we also report the cumulative reward of the agent. At each timestep the reward [46] penalises the agent for deviating from the lane center, for driving too slowly/fast, or for causing infractions. It measures how well the agent drives at the timestep level. In order to account for the length of the simulation (due to various stochastic events, it can be longer or shorter), we also report the normalised cumulative reward. More details on the experimental setting is given in Appendix D.

# 5 Results

## 5.1 Driving Performance

We evaluate our model inside the CARLA simulator on a town and weather conditions never seen during training. We picked Town05 as it is the most complex testing town, and use the 10 routes of Town05 as specified in the CARLA challenge [45], in four different weather conditions. Table 1 shows the comparison against prior state-of-the-art methods: CILRS [17], LBC [47], TransFuser [48], Roach [9], and LAV [10]. We evaluate these methods using their publicly available pre-trained weights.

MILE outperforms previous works on all metrics, with a 31% relative improvement in driving score with respect to LAV. Even though some methods have access to additional sensor information such as LiDAR (TransFuser [48], LAV [10]), our approach demonstates superior performance while only

Table 2: Ablation studies. We report driving performance on a new town and new weather conditions in CARLA. Results are averaged across three runs.

| | Driving Score | Route | Infraction | Reward | Norm. Reward |
|---|---|---|---|---|---|
| Single frame, no 3D | $51.8 \pm 3.0$ | $78.3 \pm 3.0$ | $68.3 \pm 2.8$ | $1878 \pm 296$ | $0.20 \pm 0.04$ |
| Single frame | $59.6 \pm 3.6$ | $94.5 \pm 0.6$ | $64.7 \pm 3.3$ | $6630 \pm 168$ | $0.60 \pm 0.01$ |
| No 3D | $63.0 \pm 1.5$ | $91.5 \pm 5.5$ | $\mathbf{69.1 \pm 2.8}$ | $4564 \pm 1791$ | $0.40 \pm 0.15$ |
| No prior/post. matching | $\mathbf{63.3 \pm 2.2}$ | $91.5 \pm 5.0$ | $68.7 \pm 1.8$ | $6084 \pm 1429$ | $0.55 \pm 0.07$ |
| No segmentation | $55.0 \pm 3.3$ | $92.5 \pm 2.4$ | $60.9 \pm 3.9$ | $7183 \pm 107$ | $0.64 \pm 0.02$ |
| **MILE** | $61.1 \pm 3.2$ | $\mathbf{97.4 \pm 0.8}$ | $63.0 \pm 3.0$ | $\mathbf{7621 \pm 460}$ | $\mathbf{0.67 \pm 0.02}$ |
| Expert | $88.4 \pm 0.9$ | $97.6 \pm 1.2$ | $90.5 \pm 1.2$ | $8694 \pm 88$ | $0.70 \pm 0.01$ |

using RGB images from the front camera. Moreover, we observe that our method almost doubles the cumulative reward of Roach (which was trained on the same dataset) and approaches the performance of the privileged expert.

## 5.2 Ablation Studies

We next examine the effect of various design decisions in our approach.

**3D geometry.** We compare our model to the following baselines. *Single frame* that predicts the action and BeV segmentation from a single image observation. *Single frame, no 3D* which is the same model but without the 3D lifting step. And finally, *No 3D* which is MILE without 3D lifting. As shown in Table 2, in both cases, there is a significant drop in performance when not modelling 3D geometry. For the single frame model, the cumulative reward drops from 6084 to 1878. For MILE, the reward goes from 7621 to 4564. These results highlights the importance of the 3D geometry inductive bias.

**Probabilistic modelling.** At any given time while driving, there exist multiple possible valid behaviours. For example, the driver can slightly adjust its speed, decide to change lane, or decide what is a safe distance to follow behind a vehicle. A deterministic driving policy cannot model these subtleties. In ambiguous situations where multiple choices are possible, it will often learn the mean behaviour, which is valid in certain situations (e.g. the mean safety distance and mean cruising speed are reasonable choices), but unsafe in others (e.g. in lane changing: the expert can change lane early, or late; the mean behaviour is to drive on the lane marking). We compare MILE with a *No prior/post. matching* baseline that does not have a Kullback-Leibler divergence loss between the prior and posterior distributions, and observe this results in a drop in cumulative reward from 7621 to 6084.

## 5.3 Fully Recurrent Inference in Closed-Loop Driving

We compare the closed-loop performance of our model with two different strategies:

(i) **Reset state**: for every new observation, we re-initialise the latent state and recompute the new state $[h_T, s_T]$, with $T$ matching the training sequence length.

(ii) **Fully recurrent**: the latent state is initialised at the beginning of the evaluation, and is recursively updated with new observations. It is never reset, and instead, the model must have learned a representation that generalises to integrating information for orders of magnitude more steps than the $T$ used during training.

Table 3 shows that our model can be deployed with recurrent updates, matching the performance of the *Reset state* approach, while being much more computationally efficient ($7\times$ faster from 6.2Hz with $T = 12$ of fixed context to 43.0Hz with a fully recurrent approach). A hypothesis that could explain why the *Fully recurrent* deployment method works well is because the world model has learned to always discard all past information and rely solely on the present input. To test this hypothesis, we add Gaussian noise to the past latent state during deployment. If the recurrent network is simply discarding all past information, its performance should not be affected. However in Table 3, we see that the cumulative reward significantly decreases, showing our model does not simply discard all past context, but actively makes use of it.

Table 3: Comparison of two deployment methods. (i) *Reset state*: for each new observation a fresh state is computed from a zero-initialised latent state using the last $T$ observations, and (ii) *Fully recurrent*: the latent state is recurrently updated with new observations. We report driving performance on an unseen town and unseen weather conditions in CARLA. Frequency is in Hertz.

|                 | Driving Score | Route        | Infraction    | Reward        | Norm. Reward  | Freq. |
|-----------------|---------------|--------------|---------------|---------------|---------------|-------|
| Reset state     | 61.1 ± 3.2    | **97.4 ± 0.8** | 63.0 ± 3.0  | **7621 ± 460** | **0.67 ± 0.02** | 6.2 |
| Fully recurrent | **62.1 ± 0.5** | 93.5 ± 4.8  | **66.6 ± 3.4** | 7532 ± 1122 | 0.67 ± 0.04   | **43.0** |
| Recurrent+noise | 48.8 ± 1.8    | 81.1 ± 7.0   | 61.5 ± 6.4    | 3603 ± 780    | 0.35 ± 0.07   | **43.0** |

## 5.4 Long Horizon, Diverse Future Predictions

Our model can imagine diverse futures in the latent space, which can be decoded to BeV semantic segmentation for interpretability. Figure 2 shows examples of multi-modal futures predicted by MILE.

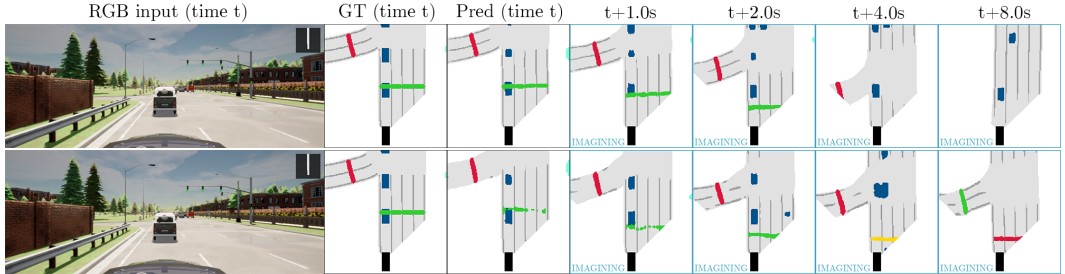

Figure 2: Qualitative example of multi-modal predictions, for 8 seconds in the future. BeV segmentation legend: black = ego-vehicle, white = background, gray = road, dark gray=lane marking, blue = vehicles, cyan = pedestrians, green/yellow/red = traffic lights. Ground truth labels (GT) outside the field-of-view of the front camera are masked out. In this example, we visualise two distinct futures predicted by the model: 1) (top row) driving through the green light, 2) (bottom row) stopping because the model imagines the traffic light turning red. Note the light transition from green, to yellow, to red, and also at the last frame $t + 8.0$s how the traffic light in the left lane turns green.

# 6 Insights from the World Model

## 6.1 Latent State Dimension

In our model, we have set the latent state to be a low-dimensional 1D vector of size 512. In dense image reconstruction however, the bottleneck feature is often a 3D spatial tensor of dimension (channel, height, width). We test whether it is possible to have a 3D tensor as a latent probabilistic state instead of a 1D vector. We change the latent state to have dimension $256 \times 12 \times 12$ (40k distributions), $128 \times 24 \times 24$ (80k distributions), and $64 \times 48 \times 48$ (160k distributions, which is the typical bottleneck size in dense image prediction). Since the latent state is now a spatial tensor, we adapt the recurrent network to be convolutional by switching the fully-connected operations with convolutions. We evaluate the model in the reset state and fully recurrent setting and report the results in Figure 3.

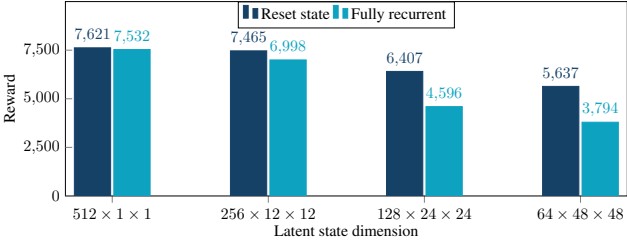

Figure 3: Analysis on the latent state dimension. We report closed-loop driving performance in a new town and new weather in CARLA.

In the reset state setting, performance decreases as the dimensionality of the latent state increases. Surprisingly, even though the latent space is larger and has more capacity, driving performance is negatively impacted. This seems to indicate that optimising the prior and posterior distributions in the latent space is difficult, and especially more so as dimensionality increases. The prior, which is a multivariate Gaussian distribution needs to match the posterior, another multivariate Gaussian distribution. What makes this optimisation tricky is that the two distributions are non-stationary and change over time during the course of training. The posterior needs to extract the relevant information from the high-resolution images and incorporate it in the latent state in order to reconstruct BeV segmentation and regress the expert action. The prior has to predict the transition that matches the distribution of the posterior.

Even more intriguing is when we look at the results in the fully recurrent deployment setting. When deployed in a fully recurrent manner in the simulator, without resetting the latent state, the model needs to discard information that is no longer relevant and continuously update its internal state with new knowledge coming from image observations. In our original latent state dimension of $512$, there is almost no different in driving performance between the two deployment modes. The picture is dramatically different when using a higher dimensional spatial latent state. For all the tested dimensions, there is a large gap between the two deployment settings. This result seems to indicate that the world model operating on high-dimensional spatial states has not optimally learned this behaviour, contrarily to the one operating on low-dimensional vector states.

## 6.2 Driving in Imagination

Humans are believed to build an internal model of the world in order to navigate in it [49, 50, 51]. Since the stream of information they perceive is often incomplete and noisy, their brains fill missing information through imagination. This explains why it is possible for them to continue driving when blinded by sunlight for example. Even if no visual observations are available for a brief moment, they can still reliably predict their next states and actions to exhibit a safe driving behaviour. We demonstrate that similarly, MILE can execute accurate driving plans entirely predicted from imagination, without having access to image observations. We qualitatively show that it can perform complex driving maneuvers such as navigating a roundabout, marking a pause a stop sign, or swerving to avoid a motorcyclist, using an imagined plan from the model (see supplementary material).

Quantitatively, we measure how accurate the predicted plans are by operating in the fully recurrent setting. We alternate between the *observing mode* where the model can see image observations, and the *imagining mode* where the model has to imagine the next states and actions, similarly to a driver that temporarily loses sight due to sun glare. In Appendix A.1 we show that our model can retain the same driving performance with up to 30% of the drive in imagining mode. This demonstrates that the model can imagine driving plans that are accurate enough for closed loop driving. Further, it shows that the latent state of the world model can seamlessly switch between the observing and imagining modes. The evolution of the latent state is predicted from imagination when observations are not available, and updated with image observations when they become accessible.

## 7 Conclusion

We presented MILE: a Model-based Imitation LEarning approach for urban driving, that jointly learns a driving policy and a world model from offline expert demonstrations alone. Our approach exploits geometric inductive biases, operates on high-dimensional visual inputs, and sets a new state-of-the-art on the CARLA simulator. MILE can predict diverse and plausible future states and actions, allowing the model to drive from a plan entirely predicted from imagination.

An open problem is how to infer the driving reward function from expert data, as this would enable explicit planning in the world model. Another exciting avenue is self-supervision in order to relax the dependency on the bird's-eye view segmentation labels. Self-supervision could fully unlock the potential of world models for real-world driving and other robotics tasks.

**Acknowledgements.** We would like to thank Vijay Badrinarayanan, Przemyslaw Mazur, and Oleg Sinavski for insightful research discussions. We are also grateful to Lorenzo Bertoni, Lloyd Russell, Juba Nait Saada, Thomas Uriot, and the anonymous reviewers for their helpful feedback and comments on the paper.

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
