# OpenReview forum: "Model-Based Imitation Learning for Urban Driving"
_NeurIPS.cc/2022/Conference — NeurIPS 2022 Accept_

### Official Review · Reviewer_W3CA · 2022-07-10

**Rating:** 6
**Confidence:** 4
**Ethics Flag:** Yes
**Soundness:** 3 good
**Presentation:** 3 good
**Contribution:** 3 good

**Summary:**

This paper proposes a predictive model of visual data for the task of imitation learning in CARLA. The model is trained to forecast RGB images, semantic segmentation images, and expert actions in the variational inference framework. The proposed method incorporates several new components, including a 3D geometry inductive bias that leverages camera intrinsics and extrinsics, as well as a variation on training called “observation dropout”. Experiments illustrate the effectiveness of the proposed approach on several towns in CARLA, relative to other recent approaches. The proposed approach is overall the most performant. Ablations also illustrate the importance of the 3D inductive bias in the single frame prediction setting, the importance of multi-frame prediction, the importance of the semantic segmentation modeling, and the importance of stochasticity.

**Questions:**

Aside from addressing the main points in the weaknesses (the two regarding the 'main claims' are the most important)

- The “deterministic temporal” baseline needs more explanation. Is it using the same architecture? What is the precise training objective?

**Limitations:**

No, there is no clear "limitations" section. The paper requires some discussion of limitations relative to (a) scalability w.r.t. availability of training data (b) feasibility w.r.t. existing model-free imitation learning methods, among others.

**Strengths And Weaknesses:**

# Strengths
- The presents the first evaluation of a visual world model on the task of offline imitation learning for urban driving in simulation
- The paper demonstrates compelling performance results of the main method
- The paper presents an interesting approach for “lifting” the representation into 3D using the camera intrinsics and extrinsics was incorporated into the model.

# Weaknesses
Overall, there are many proposed components and claims of the method that lack quantitative evidence.

- In the main claims of the introduction, L35 claims that the method “scales … by leveraging 3D geometry as an inductive bias”. However, the 3D inductive bias was not ablated from the main method (only from the single-frame prediction). Therefore, the claim has no evidence. Quantitative proof of this claim is needed in order me to agree with the inclusion of this claim, and thus I must weaken my evaluation of the paper's soundness and contributions.
- In the main claims of the introduction, L59 claims that “Our model predicts a distribution of diverse and plausible futures that can be decoded”. However, there is no quantitative evaluation on the diversity or the plausibility of the forecasted futures. This claim also appears in the conclusion. Quantitative proof of this claim is needed in order me to agree with the inclusion of this claim, and thus I must weaken my evaluation of the paper's soundness and contributions.
- A missing point of comparison is an existing model-based imitation learning approach that has been applied to CARLA [A]. This approach models only the future positional trajectory of the ego-vehicle. Although it would require a bit of experimentation, it would be informative to compare against it, or something like it, that purely models the future motion of the ego-vehicle, as opposed to the entire image. I suspect the current approach might perform quite well just by modeling the future positions, as opposed to the future semantic segmentation and RGB observations. I would at least like to see the proposed approach evaluated against itself when just trained to forecast the future positions of the vehicle.
- L100 The high resolution input is quite interesting, but it was not ablated (relative to the standard Dreamer model, which uses 64x64x3 images). The evaluation would be improved if a comparison to simply resizing the larger dimension of the input image to be 64 and running that as a comparison. This would enable us to understand the effect of using a higher resolution image. It is possible there is no effect, or that there is a significant effect. The results are important and relevant in either case.
- “Observation dropout” in S3.5 was a new method that was proposed but not ablated. The proposed approach should ablate this proposed component in order to highlight its efficacy.
- The ablation of the semantic segmentation targets for the main method is missing. The conclusion mentions “another exciting avenue is self-supervision in order to relax the dependency on the bird’s-eye view segmentation labels”. However, this should have been already investigated by simply removing this supervision from the main model and investigating its effect.

## Post-rebuttal update
The weaknesses I described were more-or-less addressed in the revised version and the authors' responses.

[A] Deep Imitative Models for Flexible Inference, Planning, and Control, ICLR 2020. https://openreview.net/forum?id=Skl4mRNYDr

---

> ### Author Response · Authors · 2022-08-02
> **Response to Review W3CA**
>
> We thank the reviewer for their thorough feedback and comments. We are pleased to see they appreciated the novelty of our visual world model for urban driving, as well as how our ablations illustrate the importance of 3D geometry, semantic segmentation, and probabilistic modelling.
>
> The reviewer’s main concern was that some claims made in the paper lacked quantitative evidence. We have updated the paper to include additional quantitative results.
>
> __3D inductive bias__
>
> We have ablated the 3D lifting from the main method, and similarly to the single frame model, this resulted in a large performance drop as shown in Table 2. The cumulative reward decreased by 41% from 7621 to 4522. This is additional evidence that modelling 3D geometry is important for urban driving.
>
> __Prediction of diverse and plausible futures__
>
> We have added an experiment in Appendix B.1 to evaluate the plausibility of the predicted states and actions from our model. We deployed the model in the simulator in the fully recurrent setting (i.e. the recurrent state is updated continuously with new observations and never reset). At regular intervals, we make our model “dream” and predict future states and actions without having access to new image observations, and execute those actions in the simulator. After this episode of dreaming, the model observes a few frames to update its knowledge of the world, and then dreams again. We repeat this procedure over the whole evaluation, with different dreaming time windows (see Appendix B.1 for more details).
> Figure 4 shows that our model can dream for up to 30% of the time without any performance degradation in either driving performance or segmentation prediction accuracy. This experiment quantitatively shows that MILE can predict plausible future states and actions.
>
> We agree with the reviewer that quantitatively evaluating whether the predicted futures are diverse is an important problem. However, this task is hard as we only have access to a single future. It could be possible to do so in simulation by randomising seeds and creating different futures from the same starting point.
>
> __Comparison with “Deep Imitative Models for Flexible Inference, Planning, and Control”__
>
> Overall the approach of [A] is different because they do not model the world/environment. They inferred future trajectories of the ego-agent from expert demonstrations, and conditioned on some specified goal to perform new tasks.
> [A] models the probability distribution of actions rather than the states of the world. Our model can jointly predict static scene, dynamic scene, and ego-actions.
>
> A direct comparison is not possible without running their model. However the public implementation runs on an older version of CARLA, which makes comparison difficult. We thank the reviewer for the reference and have added this method to the related works.
>
> __Image resolution__
>
> We evaluated the importance of image resolution by training MILE at different resolutions: 75x120, 150x240, 300x480, and 600x960 (our proposed resolution). We reported the results in Appendix B.2 Table 4 and observed a significant decrease in both driving score and cumulative reward. The performance drop is most severe in the infraction penalty metric. To get a better understanding of what is happening, we detailed in Table 5 the breakdown of the infractions. We reported the number of red lights run, the number of vehicle collisions, and the number of pedestrian collisions, all per kilometre driven. As the resolution of the image lowers, the number of infractions increases across all modalities (red lights, vehicles, and pedestrians). These results highlight the importance of high resolution images to reliably detect traffic lights, vehicles, and pedestrians. Figure 7 and 8 illustrate how traffic lights and pedestrians become much harder to distinguish in lower image resolutions.
>
> __Observation dropout and semantic segmentation labels__
>
> We have ablated the observation dropout and semantic segmentation labels in Table 2, resulting in, respectively, a 29% and 7% decrease in cumulative reward.
>
> __Deterministic temporal baseline__
>
> In this baseline, we have removed the probabilistic modelling. There is no longer a prior and posterior networks and a Kullback-Leibler divergence loss to match the two distributions. The temporal information is aggregated through the same recurrent neural network f. The purpose of this experiment was to investigate the effect of probabilistic modelling.

---

> > ### Comment · Reviewer_W3CA · 2022-08-08
> > **Response, update appendix?**
> >
> > Thank you for your response. I respond to your responses below:
> >
> > - *3D inductive bias*: The inclusion of evidence for the geometry claim is good.
> > - *Prediction of diverse and plausible futures*: From what I can tell, the Appendix has not been updated. I see no Appendix Section B or B.1
> > - *Comparison with “Deep Imitative Models for Flexible Inference, Planning, and Control”*: The comparison that would have satisfied my concern on this point is something that models _only_ the future actions (or positions) of the ego-agent in a model-based way. From what I can tell, there's no ablation of the high-dimensional observation targets (both $\hat o$ and $\hat y$ simultaneously) from the learning objective. The main difference between [A] and the proposed paper is that the learning objective includes the high-dimensional predictions, basically, both are "Model-Based Imitation Learning for Urban Driving", but the submitted paper presents a model of high-dimensional predictions, rather than just the low-dimensional information. Therefore, it would be very informative to run that ablation.
> > - *Image resolution*: From what I can tell, the Appendix has not been updated. I see no Appendix Section B or B.2 Table 4.
> > - *Deterministic temporal baseline*: Please include the exact learning objective for this baseline somewhere in the paper. It's unclear without it.

---

> > > ### Author Response · Authors · 2022-08-08
> > > **Response to Review W3CA**
> > >
> > > Thank you for the additional comments. The appendix has been updated in the 02/08 revision in the submission history. From the discussion above, it seems like Reviewer FfUk could see Appendix B. However if the 02/08 version still doesn’t appear for you we can try to upload the paper again, but we’re not sure whether we can do that during the discussion period ?
> > >
> > > - The section about __"Prediction of diverse and plausible futures"__ is on page 19-21 Appendix B.1, and the section about __"Image resolution"__ is on page 22-23 Appendix B.2.
> > > - __Comparison with “Deep Imitative Models for Flexible Inference, Planning, and Control”.__ That makes sense. Actually, since we set the image reconstruction loss to zero in our experiments (as specified in Appendix A.2 L531), the added “No segmentation” row in Table 2 is exactly what you are describing. This ablation only regresses future actions and does not reconstruct to high dimensional bird’s-eye view segmentation. We observe a decrease in both cumulative reward (7621 -> 7085) and driving score (61.1 -> 53.6). This result seems to indicate that modelling high-dimensional observations improves driving performance.
> > > - __Deterministic temporal baseline__: we will include a description of this baseline in the paper.

---

> > > > ### Comment · Reviewer_W3CA · 2022-08-08
> > > > **Response**
> > > >
> > > > Thank you, I am able to see the revisions at the end of the main paper. I looked in the supplementary .zip for the appendix, as that's where I found it previously. After considering these revisions altogether, I believe the weaknesses I stated were mostly addressed, although a few things should still be improved. I will increase my score to weak accept.
> > > >
> > > > - **"Prediction of diverse and plausible futures and Image resolution**. The image ablations are included, thank you, although the appendix section doesn't describe the scale of evaluation (how many new towns were used?). The open-loop prediction results present quantitative evidence of the degree of plausibility of the predicted rollouts w.r.t. the action quality and the segmentation forecasting quality. As noted in your response, a measure of the diversity is more difficult. One potential metric is the average log-likelihood of different potential futures in situations known to have diverse outcomes from the expert (e.g. running the expert with different goals at an intersection with the same dynamic and static configuration). It's okay that this evidence is missing, it just means that the paper should replace "diverse" with "qualitatively diverse", altogether remove it, or dedicate a few sentences to describing the difficulties with evaluating the diversity evaluation and caveating the claims that they are qualitative.
> > > > - **Comparison**, OK, I now understand that no high-dimensional reconstruction loss was used in that experiment. However, it should be clear from Table 2 alone that that is true (instead of a small clarification in the appendix), because that's quite a different experiment (predicting only the actions is quite different from predicting the actions and the images). Perhaps change it to "only actions" instead of "no segmentation".

---

> > > > > ### Author Response · Authors · 2022-08-09
> > > > > **Response to Review W3CA**
> > > > >
> > > > > Yes the appendix was originally in the supplementary.zip but we have included the updated Appendix directly in the main paper so that it was easier for reviewers to have access to all the modifications in a single document. Sorry for the confusion this has created.
> > > > >
> > > > > - __"Prediction of diverse and plausible futures and Image resolution."__ As stated in L251 we have tested our models (including in the experiments on image resolution) in the new town 05 as it is the most complex town. We will make that clearer in the captions of Table 4 and 5. Thank you for pointing us to the average log-likelihood to measure diversity of the predicted futures. We will include a few sentences to describe the difficulty of quantitatively evaluating prediction diversity, and specify that we only qualitatively show the diversity of predicted futures.
> > > > > - __"Comparison."__ That is a good point, we will change the experiment name from "no segmentation" to "only actions".
> > > > >
> > > > > Thank you for this discussion, we believe that the additional experiments and insights helped us improve the paper.

---

### Official Review · Reviewer_s7YQ · 2022-07-11

**Rating:** 7
**Confidence:** 3
**Soundness:** 3 good
**Presentation:** 3 good
**Contribution:** 3 good

**Summary:**

The paper presents an offline imitation learning approach that simultaneously predicts the evolution of the environment and imitates the expert driver. The proposed approach uses camera-based observations and a semantic segmentation map to build a world representation. The camera images are lifted to 3D and combined into a BeV representation. The environment evolution is performed in a compact latent space. The proposed method achieves better performance and generalization than prior work on a CARLA benchmark.

**Questions:**

In addition to addressing the above listed weaknesses, I have the following questions for the authors. Assuming these questions and concerns are addressed, I am currently inclined to raise the score to an accept.

1. How would the BeV semantic segmentation labels be achieved without LiDAR in a non-simulated setting?
2. What is the explicit definition of the lift operation in line 157?
3. In lines 209 and 210, it is stated that the RL expert was trained using privileged information (e.g., BeV semantic segmentation). However, my understanding was that the proposed model is also trained on BeV semantic segmentation. What is the privileged information available to the RL expert, but not the proposed model?
4. Are any of the considered baselines RL agents outside the expert? One of the motivations behind the proposed method is the ability to avoid constructing a reward. However, the cumulative reward is a key metric of evaluation for the proposed method, thus a comparison to an RL agent trained with this reward while having access to the same input information as MILE would be worthwhile.
5. Is higher better for the infraction penalty?
6. What makes the recurrent approach faster in Sec. 5.4? Were the models for the two evaluation methods trained differently?

**Limitations:**

The authors have listed some limitations as part of future work. Some potential additional limitations to include are: imitation learning methods can be subject to adversarial attacks and the model does not have safety guarantees.

**Strengths And Weaknesses:**

Strengths:
* The paper is well written and the ideas are easy to follow.
* The problem is well-motivated and the literature review does a good job at contextualizing the paper in prior work.
* The idea behind end-to-end feature learning, while incorporating geometric bias is intuitive.
* Overall, the empirical evaluation of the method is extensive and convincing. The analysis and discussion is thoughtfully constructed, and the different components of the architecture were thoroughly ablated.
* The figures are informative and effectively illustrate the proposed approach and results.

Weaknesses:
* The sentence in the abstract is a bit strong 'So far, such world models have been shown to be highly effective at solving games, but only in simple visual environments with little interaction among agents.'. There has been lots of work in the trajectory prediction setting that builds world models in dynamic environments (e.g., [1,2]).
* The related works section is missing a discussion of trajectory prediction methods which are closely related to imitation learning as presented in the paper.
* One of the motivations for the imitation learning approach over RL is the difficulty in constructing rewards. However, in the evaluation, the most effective metric is determined to be cumulative reward. The considered baselines do not seem to include an RL approach with access to the same input information as MILE.

[1] Zhao, Hang, et al. "TNT: Target-driven Trajectory Prediction." Conference on Robot Learning. PMLR, 2021.
[2] Salzmann, Tim, et al. "Trajectron++: Dynamically-feasible trajectory forecasting with heterogeneous data." European Conference on Computer Vision. Springer, Cham, 2020.

The paper needs to be proofread for typos. The following is a non-exhaustive list of the typos I found:

1. The BeV acronym is defined in line 30, but not consistently used after that.
2. Line 106: 'a the full description' should read 'the full description'.
3. Line 168: 'indicating the agent where to navigate' should read 'indicating to the agent where to navigate'.
4. Line 234: 'as it obtained' should read 'as it is obtained'.
5. The references should be proofread (e.g., to ensure the year is not entered twice in a citation, the conference venue is listed instead of ArXiv when available, etc.).

---

> ### Author Response · Authors · 2022-08-02
> **Response to Review s7YQ**
>
> We thank the reviewer for their insightful feedback and comments. We are pleased to see they found the idea of learning a world model while incorporating 3D geometry well-motivated and intuitive. We are glad they found the empirical evaluation, analysis and discussion thoughtfully constructed and convincing.
>
> __Trajectory prediction, imitation learning and world models__
>
> _“There has been lots of work in the trajectory prediction setting that builds world models in dynamic environments (e.g., [1,2]).”_
>
> Trajectory forecasting aims at estimating the future trajectories of dynamic agents using past physical states (e.g. position, velocity), and scene context (e.g. as an offline HD map). World models build a latent representation of the environment that explains the observations from the sensory inputs of the ego-agent (e.g. camera images) conditioned on their actions. While trajectory forecasting methods only model the dynamic scene, world models jointly reason on static and dynamic scenes. The future trajectories of moving agents is implicitly encoded in the learned latent representation of the world model, and could be explicitly decoded given we have access to future trajectory labels.
>
> More importantly, trajectory forecasting methods assume access to past physical states of dynamic agents, and to scene context through an offline HD map. World models have to infer these quantities from image observations only, in order to accurately predict future states.
> For these reasons, even though [1,2] model the dynamic environment, they cannot be considered as world models.
>
>
> _“The related works section is missing a discussion of trajectory prediction methods which are closely related to imitation learning as presented in the paper.”_
>
> We have added a section in the related works to discuss how trajectory forecasting methods are related to imitation learning and world models.
>
>
> __Comparison with an RL baseline__
>
> _“The considered baselines do not seem to include an RL approach with access to the same input information as MILE.”_
>
>
> To our knowledge, [3] is the only camera-based RL model evaluated on CARLA. We could not compare our model with their method as their public code repository did not have the training script, but only the inference script of a model trained on an older version of CARLA.
>
> __Typos__
>
> We thank the reviewer for pointing out the typos, which are now corrected. We have also fully proofread the paper.
>
> __Questions__
>
> 1. Labels in non-simulated settings could be obtained using a combination of manual labelling and labels from a teacher model, such as BEVFormer [4] that recently won first prize in the “Waymo Open Dataset challenge”. Teacher labels will be of high quality but not as perfect as the ones obtained from the simulator.
> It would be possible to evaluate the effect of replacing perfect simulator labels with imperfect labels from a teacher model the following way. (i) We train a teacher BeV model on a small subset of annotated data, (ii) we label the whole dataset with the teacher, (iii) we train MILE using the teacher labels and evaluate the impact on performance.
> 2. We have included a description of the “lift” operation in Appendix A.2.
> 3. The RL expert used to collect expert demonstrations was trained using privileged BeV segmentation as an input to the model. Our proposed model does not assume access to BeV segmentation during inference, but only to camera inputs. The BeV labels are only used during training.
> 4. See paragraph above “Comparison with an RL baseline”.
> 5. Yes, higher is better for the infraction penalty. At the beginning of an evaluation run, the infraction penalty is set to 100. A multiplicative penalty is applied for each infraction. For example hitting two cars would result in an infraction penalty of 100*0.6^2=36.
> 6. The exact same model was used in the two evaluation methods (reset state and fully recurrent). The difference is that in “reset state”, every time a new observation comes in, the hidden state of the world model is reset, and the whole past sequence of observations (o_1,...,o_T) has to be processed to compute the new latent state (h_T, s_T).
> In the fully recurrent approach, the hidden state of the world model is never reset, but continuously updated in a recursive manner for each new observation. For example if a new frame o_{t+1} is observed, then the latent state [h_t, s_t] is updated using the deterministic and stochastic updates defined in L135-136.
> The difference in inference speed comes from the fact that only a single observation is processed in the “fully recurrent” approach, whereas the whole sequence of past observations has to be processed in the “reset state” approach.
>
>
> [3] “End-to-End Model-Free Reinforcement Learning for Urban Driving using Implicit Affordances”, Toromanoff et al., CVPR 2020.
>
> [4] “BEVFormer: Learning Bird's-Eye-View Representation from Multi-Camera Images via Spatiotemporal Transformers”, Li et al., ECCV 2022.

---

> > ### Comment · Reviewer_s7YQ · 2022-08-03
> > **Response to Authors**
> >
> > Thank you for the clarifications and the related works additions! Below are some of my follow-up thoughts.
> >
> > > "While trajectory forecasting methods only model the dynamic scene, world models jointly reason on static and dynamic scenes."
> >
> > This reasoning makes sense. In this case, I think it would be pertinent to discuss the environment prediction line of work (e.g., [1-6]) that make predictions and reason about both the static and dynamic environments.
> >
> > [1] Itkina, Masha, Katherine Driggs-Campbell, and Mykel J. Kochenderfer. "Dynamic environment prediction in urban scenes using recurrent representation learning." 2019 IEEE Intelligent Transportation Systems Conference (ITSC). IEEE, 2019.
> >
> > [2] Toyungyernsub, Maneekwan, et al. "Double-prong ConvLSTM for spatiotemporal occupancy prediction in dynamic environments." 2021 IEEE International Conference on Robotics and Automation (ICRA). IEEE, 2021.
> >
> > [3] Lange, Bernard, Masha Itkina, and Mykel J. Kochenderfer. "Attention Augmented ConvLSTM for Environment Prediction." 2021 IEEE/RSJ International Conference on Intelligent Robots and Systems (IROS). IEEE, 2021.
> >
> > [4] Mohajerin, Nima, and Mohsen Rohani. "Multi-step prediction of occupancy grid maps with recurrent neural networks." Proceedings of the IEEE/CVF Conference on Computer Vision and Pattern Recognition. 2019.
> >
> > [5] Thomas, Hugues, et al. "Learning Spatiotemporal Occupancy Grid Maps for Lifelong Navigation in Dynamic Scenes." 2022 International Conference on Robotics and Automation (ICRA). IEEE, 2022.
> >
> > [6] Mahjourian, Reza, et al. "Occupancy flow fields for motion forecasting in autonomous driving." IEEE Robotics and Automation Letters 7.2 (2022): 5639-5646.
> >
> > 1. This is good insight, thank you!
> > 2. Thank you!
> > 3. Does the RL expert use BeV segmentations during inference?
> > 4. Noted.
> > 5. This would be good to highlight in the paper for clarity.
> > 6. Thank you for the clarification!

---

> > > ### Author Response · Authors · 2022-08-04
> > > **Response to Review s7YQ**
> > >
> > > We thank the reviewer for pointing us to the environment prediction line of work [1-6]. We will include these in the related works.
> > >
> > > __[1-4]__ predict future static and dynamic occupancy grid maps (OGMs) from past OGMs in urban driving scenes. They frame environment prediction as a video prediction problem. The static and dynamic OGMs are obtained by processing LiDAR measurements and 3D object detections.
> > >
> > > Although the output space between [1-4] and our proposed approach is similar (BeV representation), there are three key differences:
> > >
> > > 1. Both the inputs and outputs of their model are OGMs. This means the static scene can be predicted perfectly by simply estimating ego-motion from past inputs. The motion prediction of dynamic agents is also made easier as the vehicles are already represented in a metric BeV space. In contrast, our proposed model operates on high-dimensional camera images and has to do all the heavy lifting to reason about 3D geometry and semantics from images in order to predict BeV outputs.
> > > 2. Their methods are deterministic, whereas our approach is probabilistic and can predict multimodal futures.
> > > 3. They do not model ego-behaviour. We jointly predict future states and actions, and demonstrate the efficacy of the learned policy in the CARLA driving simulator.
> > >
> > >
> > > __[5]__ also predict future static and dynamic OGMs but use 3D LiDAR point clouds as inputs. They show preliminary results of a non-learned navigation system that can control a robot in a simulated Gazebo environment to avoid obstacles using the predicted OGMs. In comparison, our proposed model can jointly reason on future states and actions probabilistically. As shown in the supplementary video, it can predict diverse and plausible future states, consistent with predicted actions.
> > >
> > > __[6]__ focus on dynamic scene modelling in dense urban driving scenes. In addition to future occupancy maps of dynamic objects, and contrary to [1-4], they also predict future flow fields. A flow field indicates the 2D BeV velocity of each moving object. This allows them to use the chain of flow predictions to trace back the identity of detected moving objects back to the present time. The inputs of their model are: (i) past agent states (position, velocity, bounding box), (ii) road structure (points sampled uniformly along the line segments and curves), and (iii) traffic light states. These input points are placed in a BeV grid and processed with an encoder-decoder architecture to output future occupancy maps and future flow fields. However, similarly to [1-4], their approach is deterministic and does not model ego-behaviour.
> > >
> > >
> > > __Questions__
> > >
> > > 3 - Yes, the RL expert uses BeV segmentation as input during inference, which is not the case of our camera-only model.
> > >
> > > 5 - We will give a more precise description of how the infraction penalty is calculated.

---

### Official Review · Reviewer_FfUk · 2022-07-12

**Rating:** 6
**Confidence:** 5
**Soundness:** 3 good
**Presentation:** 3 good
**Contribution:** 3 good

**Summary:**

This work develops an model-based imitation learning pipeline for autonomous driving.
Their observation network encodes multi-view camera images and route information into a latent embedding which is then used in a variational generative model to predict future BEV semantic segmentation and expert actions.
To my knowledge, this is the first work to show the effectiveness of world models in autonomous driving simulators.

**Questions:**

* Does this work have a submission to https://leaderboard.carla.org/leaderboard/? This is important to my final rating.
* Related to the trajectory forecasting point - is there any particular reason why rasterized BEV labels were used for future prediction rather than detections (i.e. boxes)?
* BEV labels would require auto-labeling or human annotations - how much driving data is needed for a "good enough" world model?

**Limitations:**

yes

**Strengths And Weaknesses:**

### Strengths
* Clear writing, technical portion + model architecture easy to understand, since this work seems like FIERY but instead predicts driving actions.
* Their model, which uses cameras only, shows significant improvements over even lidar-based methods.
* Supplementary video shows fairly good short-term predictions.
* I agree with and am glad to see the discussion of driving score - the score is definitely biased towards certain types of routes and adding the reward/normalized reward as another source of quantitative evaluation is useful.

### Weaknesses
* The architecture figure looks more complicated than the architecture actually is. I would suggest adding a simpler toy figure in place of Figure 1, moving the current figure to the technical section.
* I would like to see a bit of discussion or related works on trajectory forecasting. It would be beneficial to discuss some of the difference between world models and trajectory forecasting since there are many recurring themes between these two areas.
* Adding dataset size to Table 1 would be helpful for comparison.
* It seems like a few important experiments refer solely to the supplementary - I'd suggest some spacing edits to try to fit these into the main paper.

---

> ### Author Response · Authors · 2022-08-02
> **Response to Review FfUk**
>
> We thank the reviewer for their suggestions and comments. We are glad they appreciated how this work showed the effectiveness of world models in urban driving simulators, as well as the quality of the predicted futures.
>
> __Trajectory forecasting and word models__
>
> _“I would like to see a bit of discussion or related works on trajectory forecasting”_
>
> Trajectory forecasting aims at predicting the future positions of dynamic agents, given their past trajectory and scene context. The methods in trajectory forecasting operate on physical states (e.g. position, velocity), and scene context is given as an offline map or inferred from LiDAR inputs [1, 2, 3, 4].
>
> World models learn latent states that model the dynamics of the environment conditioned on the actions of the ego-agent. Both the static scene and the dynamic scene are modelled. The world model does not have access to past physical states and scene context in the form of an offline map, but only to observations from the sensory inputs of the ego-actor. It has to learn latent dynamics that explain the observations of the ego-agent.
>
> The static scene and motion of dynamic agents is thus modelled implicitly in the latent state, but can be decoded into interpretable outputs (e.g. bird’s-eye view semantic segmentation, future trajectories of agents). Joint modelling of static scene and dynamic scene can be helpful for planning and acting. For instance, interactions between traffic lights and other dynamic agents are better modelled jointly.
>
> We have added a section in the related works to discuss how trajectory forecasting methods are related to world models.
>
>
>
>
> __Comments__
>
> - _“Adding dataset size to Table 1 would be helpful for comparison.”_
>
> We have added the dataset size of prior works in Appendix Table 6. We found that most recent methods (including ours) have a similar dataset size of around 20-30 hours of driving data.
> - _“It seems like a few important experiments refer solely to the supplementary”_
>
> We have edited the paper to include more experiments from the supplementary in the main paper.
> - _“Does this work have a submission to https://leaderboard.carla.org/leaderboard/?”_
>
> This work does not yet have a public submission on the leaderboard, but it is a task we are actively working on.
>
> __On BeV labels instead of 3D bounding boxes__
>
> _“Is there any particular reason why rasterized BEV labels were used for future prediction rather than detections (i.e. boxes)?”_
>
> - If the decoder to BeV segmentation was replaced with a decoder to 3D bounding boxes, the static and dynamic components would be decoupled (assuming there is an additional decoder for static scene). This decoupling is not advantageous since the dynamic agents’ actions are conditioned on the static scene in which they operate.
> - We chose to represent the scene in the ground plane because driving happens on the 2D road, which is an effective inductive bias for the network.
> - BeV segmentation makes the architecture simpler since it allows us to not limit the maximum number of agents in the scene and not use a recurrent-style decoder from the latent state. Instead, we simply use a convolutional network to decode to BeV.
>
> __How much data is needed?__
>
> _“How much driving data is needed for a "good enough" world model?”_
>
> We investigated how performance scales with data. We trained our proposed model with a fraction of the full dataset (1/2, 1/4, 1/8, 1/16 and 1/32) and reported the results in Appendix Table 7. We observe similar performance across all metrics for a dataset size of 8 to 32 hours (which is the full training set). Performance however degrades monotonically starting from 4 hours of training data. We conclude that on the CARLA simulator, around 8 hours of data collected by our expert driver is sufficient.
>
>
> [1] “DESIRE: Distant Future Prediction in Dynamic Scenes with Interacting Agents”, Lee et al., CVPR 2017.
>
> [2] “PRECOG: PREdiction Conditioned On Goals in Visual Multi-Agent Settings”, Rhinehart et al., ICCV 2019.
>
> [3] "TNT: Target-driveN Trajectory Prediction", Zhao et al., CoRL 2020.
>
> [4] "Trajectron++: Dynamically-Feasible Trajectory Forecasting with Heterogeneous Data", Salzmann et al., ECCV 2020.

---

> > ### Comment · Reviewer_FfUk · 2022-08-04
> > **Response to Response to Review**
> >
> > ### Related Works
> > Paragraph looks good
> >
> > ### No Leaderboard Submission
> > Thanks for clarifying, I understand the challenges of submitting to the leaderboard.
> > Unfortunately this makes me a bit worried that if future works want to compare with MILE, they'll have to reproduce your experimental setting.
> >
> > ### Supplementary
> > * Table A.2.3 - table overflowing
> > * Table 7 (Section B.3) - why is the route score for 8h of training data "1.0" and bolded?

---

> > > ### Author Response · Authors · 2022-08-05
> > > **Response to Review FfUk**
> > >
> > > __Evaluation method__
> > >
> > > Our evaluation method is actually not new and has been used by recent works [5-7]. It consists of evaluating the driving agent on a town never seen during training and in new weather conditions. Similarly to [5-7], the routes in the evaluation are those specified by the CARLA challenge (https://leaderboard.carla.org/get_started/). There are also advantages in this evaluation method compared to the public leaderboard:
> > >
> > > - It is possible to test generalisation capabilities. We can evaluate the model in a town never seen during training and weather conditions never seen during training. In the public leaderboard, it is standard practice to train the model on all the towns and weathers the model will be evaluated on.
> > > - We can have full control over the training data and sensor inputs of the model for fair comparison. Iteration speed is also much faster - submissions in the leaderboard take 80-300 hours to complete and are limited to a single model at a time (with a compute budget of 200 hours per month). In our evaluation setting, there are no restrictions in the number of models we can evaluate, and a single evaluation takes under 12 hours on a GTX 1080Ti.
> > > - The data collection, model training, and evaluation scripts are all available in the supplementary material, and we plan to publicly release the code. It will therefore be possible for researchers who would like to build up on this work or compare their method to ours to do so.
> > >
> > > __Supplementary__
> > >
> > > - Table A.2.3. was indeed overflowing, thank you for pointing us to this.
> > >
> > > - In Table 7 (Section B.3), the route score for 8h of training data is actually 100.0. The “1.0” was a typo due to the fact that the simulator returns the route score and infraction penalty in decimal values ([0, 1]), but it is standard practice to report these metrics in [0, 100] by multiplying by 100 for readability.
> > >
> > > [5] “Multi-Modal Fusion Transformer for End-to-End Autonomous Driving”, Prakash et al., CVPR 2021.
> > >
> > > [6] “End-to-End Urban Driving by Imitating a Reinforcement Learning Coach”, Zhang et al., ICCV 2021.
> > >
> > > [7] “Learning from All Vehicles”, Chen and Krähenbühl, CVPR 2022.

---

### Meta-Review · Area_Chair_QBV2 · 2022-08-22

**Recommendation:** Accept
**Confidence:** Certain

**Metareview:**

This work introduced a model-based framework for offline imitation learning of autonomous driving policies in simulated urban environments. The proposed model MILE jointly learns a world model and predicts expert actions using a variational generative model. This paper was reviewed by three expert reviewers. At the initial reviews, the reviewers raised several questions about technical details and gave valuable suggestions on the overall presentation of the paper. In particular, Reviewer W3CA pointed out that some of the claims made in the paper were not sufficiently supported by quantitative evidence, and Reviewer s7YQ suggested an additional discussion of trajectory prediction methods. The authors did a good job drafting a detailed rebuttal and updating the paper revision, which addressed most of the reviewers' concerns. In the end, all three reviewers leaned towards accepting this paper. The AC read the paper, the reviews, and the discussions in detail and believed that this paper had presented a strong showcase of using model-based approaches for challenging vision-based autonomous driving problems. Taking all these into account, the AC recommends accepting this paper at NeurIPS 2022.

**Award:**

No

---

### Decision · Program_Chairs · 2022-09-14

Accept